# Improved Spectrum Coexistence in 2.4 GHz ISM Band Using Optimized Chaotic Frequency Hopping for Wi-Fi and Bluetooth Signals

**DOI:** 10.3390/s23115183

**Published:** 2023-05-30

**Authors:** Ashraf A. Eltholth

**Affiliations:** National Telecommunication Institute, Cairo 11765, Egypt; ashraf.elthoth@nti.sci.eg

**Keywords:** spectrum coexistence, frequency hopping, Wi-Fi, BLE, ISM band, logistic map

## Abstract

Efficiently managing coexistence is crucial for achieving high-quality wireless communication in the Industrial, Scientific, and Medical (ISM) band where multiple wireless communication systems operate. Coexistence problems between Wi-Fi and Bluetooth Low Energy (BLE) signals are especially significant due to their shared frequency band, which often leads to interference and a reduced performance for both systems. Therefore, effective coexistence management strategies are essential for ensuring the optimal performance of Wi-Fi and Bluetooth signals in the ISM band. In this paper, the authors conducted a study to investigate coexistence management in the ISM band by evaluating four frequency hopping techniques: random, chaotic, adaptive, and an optimized chaotic technique proposed by the authors. The optimized chaotic technique aimed to minimize interference and ensure zero self-interference among hopping BLE nodes by optimizing the update coefficient. Simulations were conducted in an environment with existing Wi-Fi signal interference and interfering Bluetooth nodes. The authors compared several performance metrics, including the total interference rate, total successful connection rate, and trial execution time for channel selection processing time. The results indicated that the proposed optimized chaotic frequency hopping technique achieved a better balance between reducing interference with Wi-Fi signals, achieving a high success rate for connecting BLE nodes, and requiring minimal trial execution time. This makes it a suitable technique for managing interference in wireless communication systems. While the proposed technique had a higher interference than the adaptive technique for small numbers of BLE nodes, for larger numbers of nodes it had a much lower interference than the adaptive technique. The proposed optimized chaotic frequency hopping technique provides a promising solution for effectively managing coexistence in the ISM band, particularly between Wi-Fi and BLE signals. It has the potential to improve the performance and quality of wireless communication systems.

## 1. Introduction

The wireless frequency spectrum has become increasingly congested due to the growing demand for wireless communication services. The coexistence of multiple wireless systems in the same frequency band can cause interference and performance degradation. Various methods have been proposed in the literature to address the coexistence issue in heterogeneous cognitive networks, including spectrum sensing, spectrum sharing, and spectrum allocation [1,2].

Cognitive radio networks require spectrum sensing to detect the presence and absence of primary users. Energy detection, cyclostationary detection, and matched filter detection are commonly used sensing techniques but they have limitations, such as a low detection accuracy, high computational complexity, and low robustness in the presence of noise and fading [3].

Spectrum sharing is another approach that has been widely studied, where available spectrum resources are shared among different wireless systems. Static and dynamic spectrum access, auction-based spectrum allocation [4], and game-theoretic approaches [5] are some of the most used spectrum sharing methods.

Spectrum allocation has been proposed as an approach to minimize interference between wireless systems in heterogeneous cognitive networks. This involves assigning specific frequency bands to different wireless systems using interference-aware or interference-free methods [6]. However, this approach is not a complete solution to the problem of interference in such networks.

Despite the proposed spectrum allocation approach, the coexistence of different wireless systems in heterogeneous cognitive networks remains a challenging problem. Further research is needed to develop more efficient and robust techniques that can address this issue [7]. The research trend in this area is focused on enhancing the utilization of spectrum resources, reducing interference, and improving the performance of wireless systems.

The shared frequency band in the unlicensed ISM frequency band of 2.4 GHz used by wireless communication systems such as Wi-Fi and Bluetooth makes them highly susceptible to interference and coexistence issues leading to performance degradation [8]. Several approaches have been developed to mitigate coexistence issues, including spectrum sharing, adaptive frequency hopping, and statistical signal transmission [1,4,8,9,10,11].

Chaotic frequency hopping and adaptive frequency hopping techniques have been proposed to improve the coexistence of Wi-Fi and Bluetooth signals. Chaotic frequency hopping introduces randomness in the hopping sequence, making it difficult for interfering signals to predict and track the system’s frequency. Adaptive frequency hopping dynamically adjusts the hopping pattern based on the current communication environment. The use of chaotic frequency hopping sequences is a promising approach that has shown potential for improving coexistence, but further research is needed to develop and evaluate new techniques for enhancing wireless communication system coexistence.

Recent studies have explored the potential of combining multiple coexistence strategies to enhance wireless communication system performance [9,12], including frequency-to-time mapping-based frequency hopping receivers [13] and cognitive radio techniques [14]. There is no single solution for enhancing coexistence in wireless communication systems in the 2.4 GHz ISM frequency band and a combination of approaches may be necessary.

This paper proposes using an optimized chaotic frequency hopping technique to address self-interference and interference with existing Wi-Fi signals in wireless communication systems. The researchers were motivated to explore chaotic frequency hopping due to its simplicity of generation and improved control over interference compared to random frequency hopping methods.

Generating a chaotic frequency hopping sequence using linear logistic maps is a relatively simple process that requires only one parameter to adjust the output of the sequence. This simplicity allows for the generation of a frequency hopping sequence that appears to be random while avoiding self-interference and interference with other wireless networks. By adjusting the “r” parameter, the frequency hopping sequence can be optimized to reduce the chances of self-interference and limit any interference with existing Wi-Fi signals. In contrast, random frequency hopping sequences require a much more complex generation process that typically involves the use of pseudo-random number generators. The use of these generators can lead to correlations between generated sequences or inconsistencies in the generated sequences, leading to self-interference or interference with other wireless networks. Therefore, the use of chaotic frequency hopping sequences offers a simpler and potentially more effective solution to interference issues in wireless communication systems.

The proposed optimized chaotic frequency hopping method is evaluated using simulations to analyze the coexistence between these two wireless systems in the ISM 2.4 GHz band. The results of this study are compared to those obtained from conventional frequency hopping and non-hopping systems. The main contributions of this paper include offering insights into the challenges of coexistence issues between wireless systems and proposing an alternative solution to mitigate mutual interference between Wi-Fi and Bluetooth signals.

Several performance metrics can be used to analyze the coexistence between Wi-Fi and Bluetooth, including channel occupancy, spectral purity, and throughput. Channel occupancy measures the fraction of time a channel is occupied by either Wi-Fi or Bluetooth and serves as an indicator of the available spectrum resources. Spectral purity measures the extent to which Wi-Fi and Bluetooth signals occupy the same frequency band, while throughput measures the amount of data transmitted per unit time. The simulation results provide a detailed analysis of these performance metrics to evaluate the effectiveness of the proposed optimized chaotic frequency hopping method for coexistence enhancement.

The main contributions of this study include providing insights into coexistence issues between wireless systems and proposing a novel approach to mitigate mutual interference between Wi-Fi and Bluetooth signals. Moreover, the proposed solution is not limited to Wi-Fi and BLE coexistence but is also applicable to other personal wireless communication systems operating on the 2.4 GHz band and using frequency hopping. This includes systems such as Zigbee, Bluetooth Classic, and Bluetooth Low Energy (BLE), among others. These systems are widely used in various applications, such as home automation, industrial control, and healthcare monitoring. The proposed solution utilizes frequency hopping to mitigate the interference from other wireless communication systems operating in the same band, which can improve the overall performance and reliability of these systems. Therefore, our proposed solution has the potential to be applied to a wide range of personal wireless communication systems operating in the 2.4 GHz band and using frequency hopping.

### 1.1. Data Collection

We conducted practical measurements to collect data from the 2.4 GHz ISM band using a spectrum analyzer. The measurements were performed in the National Telecommunication Institute building in Cairo, Egypt over 5 working days, resulting in 40,000 data instances for the 40 channels in the 2.4 GHz ISM band.

The data collection step aimed to encompass a diverse range of wireless network conditions to realistically model interference and self-interference. We also performed a data preparation step using the Gaussian mixed model approach to estimate the noise threshold and make a decision on the availability of each channel. This helped ensure that the data used in our simulations accurately reflected the wireless environment in which the proposed technique would be applied.

The use of a spectrum analyzer enabled us to measure the signal strength of existing Wi-Fi signals and identify potential sources of interference. We also recorded the duration of each detected signal, enabling the accurate characterization of the wireless environment’s activity over time. The data collection and preparation are depicted in Algorithm 1.
**Algorithm 1 ** Data Collection and Preparation Algorithm.
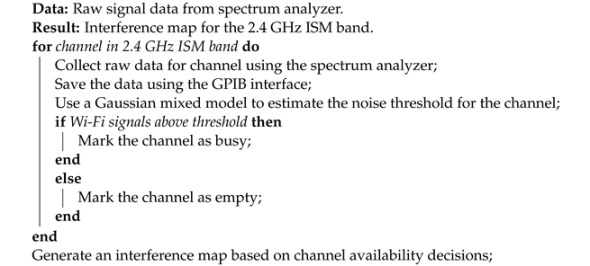


The above algorithm summarizes the key steps involved in our data collection and preparation processes. We collected 40,000 data instances for the 40 channels in the 2.4 GHz ISM band using a spectrum analyzer and recorded the collected data using the GPIB interface. We then analyzed the collected data to determine the noise threshold for each channel using a Gaussian mixed model approach, ensuring a high accuracy. Based on the estimated noise threshold, we determined the channel availability, with a channel being considered busy or empty depending on the presence or absence of Wi-Fi signals above the estimated noise threshold. We generated an interference map based on these channel availability decisions, providing an estimate of the interference experienced in the 2.4 GHz ISM band and enabling the simulation of practical applications.

Overall, the data collection step provided a rich set of data, which was utilized in our simulation study to realistically model the wireless environment. We believe that the use of practical measurements and GPIB data storage ensures the accuracy and reliability of our dataset, providing a valuable foundation for the simulation study’s subsequent data preparation and analysis steps.

Therefore, by basing the simulations on actual measurements of existing Wi-Fi signals and using a well-established statistical approach to evaluate the decision threshold, the proposed optimized chaotic frequency hopping technique became more promising. These measures helped to establish a more reliable basis for evaluating the effectiveness of the proposed technique in practical applications.

### 1.2. Research Methodology

The system model that includes four tested BLE frequency hopping techniques, three performance metrics for each technique, and the proposed optimized chaotic technique is depicted in Figure 1.

The research methodology was based on measuring the signal power for five working days using a spectrum analyzer and estimating the noise threshold using a Gaussian mixed model. An existing interference map was then produced based on the threshold and four BLE frequency hopping techniques were simulated. For each implemented technique, the execution time was tested, as well as the Wi-Fi and BLE self-interference rates, and successfully connected the BLE nodes rates. The performance metrics for each frequency hopping technique were then compared based on these factors. Finally, an optimized chaotic frequency hopping technique was proposed based on the results of the performance comparison, and the selection among the tested techniques was made based on desired performance criteria. By including multiple frequency hopping techniques and testing them across three relevant metrics, this methodology ensured that the proposed optimized technique was well-informed and thoroughly tested against other frequency hopping techniques.

## 2. Bluetooth and Wi-Fi Coexistence

Bluetooth and Wi-Fi are popular wireless technologies used in consumer electronics that share the 2.4 GHz ISM band for RF transmission. However, they differ in their transmission methods, with Bluetooth using a slotted transmission approach for power efficiency while Wi-Fi continuously transmits. This creates interference problems as Wi-Fi signals look like noise to Bluetooth receivers, and vice versa [15].

IoT gateways and hubs can benefit from supporting multiple wireless protocols, such as Wi-Fi, Zigbee, Thread, and Bluetooth, to accommodate a diverse range of end nodes. However, since these protocols use the same 2.4 GHz ISM band, performance degradation can occur when multiple radios operate simultaneously. Wi-Fi coexistence strategies can help mitigate these issues and minimize the interference between collocated radios.

Wi-Fi coexistence enables multiple 2.4 GHz technologies, including Wi-Fi, Zigbee, Thread, and Bluetooth, to operate simultaneously without interfering with each other’s signals. Interference can lead to a reduced device responsiveness and increased power consumption due to message failures and retries. However, achieving coexistence is more challenging with current gateways and hubs, which support a higher throughput, transmit at higher power levels, and integrate up to 4 2.4 GHz radios, requiring additional steps to ensure a reliable wireless performance.

The 2.4 GHz ISM band frequency plan is illustrated in Figure 2, with Wi-Fi operating on 3 non-overlapping 20 MHz channels, and Bluetooth operating on up to 40 BLE channels with 2 MHz spacing.

A Wi-Fi channel is 20 MHz wide, while a BLE channel is 2 MHz wide, allowing approximately 10 BLE channels to fit into the range of a Wi-Fi channel. Of the 40 Bluetooth channels available for the Frequency Hopping Spread Spectrum (FHSS), 10 (25%) appear within the same frequency space as a given Wi-Fi channel, as depicted in Figure 2.

For example, in a situation where a device is using collocated Wi-Fi and Bluetooth with both active simultaneously, such as making a Wi-Fi phone call while using a Bluetooth headset, a Bluetooth channel hopping into the middle of a transmitting Wi-Fi packet can corrupt it. After a few such instances, the Wi-Fi transmitter will back off to a lower speed, resulting in more time to deliver a complete packet and a decreased throughput. The probability of another collision with that packet is 25%. Should that collision happen, the Wi-Fi will further reduce the throughput. Similarly, if the Wi-Fi signal corrupts the Bluetooth packet, Bluetooth will hop to the next channel and try again if the Bluetooth transmission is asynchronous, resulting in a reduced throughput.

The coexistence of Wi-Fi and Bluetooth in the 2.4 GHz ISM band requires careful consideration and implementation of strategies to minimize interference and maximize wireless performance [12].

### 2.1. The Wi-Fi Physical Layer (PHY)

The 802.11a/g/n/ac Wi-Fi signal model is commonly employed in contemporary Wi-Fi systems. It utilizes Orthogonal Frequency Division Multiplexing (OFDM) modulation, which divides the frequency band into multiple subcarriers and applies distinct modulation schemes and coding rates for each subcarrier [16]. The 802.11a/g/n/ac model comprises various parameters, such as:Carrier frequency: This is the center frequency of the Wi-Fi signal, typically in the range of 2.4 GHz to 5 GHz.Channel bandwidth: This is the width of the frequency band used by the Wi-Fi signal, typically 20 MHz, 40 MHz, or 80 MHz.Subcarrier spacing: This is the frequency spacing between adjacent subcarriers, typically 312.5 kHz.Modulation scheme: This determines how data are mapped onto the subcarriers, typically using Quadrature Amplitude Modulation (QAM) with different constellation sizes (e.g., 16-QAM, 64-QAM, 256-QAM).Coding rate: This determines how much redundancy is added to the data to improve error correction, typically using convolutional or Reed-Solomon codes with different code rates (e.g., 1/2, 3/4, 5/6).Guard interval: This is a time interval inserted between adjacent OFDM symbols to prevent inter-symbol interference, typically 800 ns or 400 ns.

Various characteristics of Wi-Fi signals, such as data rates, susceptibility to noise and interference, and resistance to channel fading and multipath propagation, can be modeled by adjusting the parameters of Wi-Fi signal models such as the 802.11a/g/n/ac model. The Wi-Fi physical layer in the 2.4 GHz band incorporates preambles and headers in its signal structure to aid in signal detection and synchronization. The preamble establishes the timing and frequency of the signal, while the header contains information about the data type being transmitted [17].

### 2.2. Bluetooth Physical Layer Signal

The 2.4 GHz physical layer of Bluetooth adopts frequency hopping spread spectrum (FHSS) technology to minimize the interference from other devices and provide high-level security. FHSS is a radio transmission technology that uses rapidly changing frequencies. In BLE, data are transmitted over 40 frequency channels in the 2.4 GHz ISM band with a 2 MHz bandwidth each. The hopping sequence is established through a master-slave communication model where the master device determines the hopping pattern, while the slave devices follow it [18].

To encode data, the Bluetooth physical layer employs a Gaussian Frequency Shift Keying (GFSK) modulation, which involves slightly shifting the frequency of the carrier signal. The amount of frequency shift is proportional to the amplitude of the data signal. Additionally, the Bluetooth physical layer uses a preamble and header similar to Wi-Fi to aid in synchronization and signal detection. The preamble is used to establish the timing and frequency of the signal, while the header carries information about the transmitted data type.

## 3. Bluetooth LE Channel Selection Algorithms

Effective channel selection is essential to ensure the quality and reliability of Bluetooth LE communication. Numerous algorithms have been proposed to tackle this challenge with the goal of optimizing communication quality and minimizing interference. During the assignment of a carrier frequency to a BLE signal, it may either collide with an existing Wi-Fi signal or be assigned to a vacant channel depending on certain selection criteria that are discussed later. Some of the vacant channels may be unoccupied despite an interference event between the BLE and Wi-Fi signals, as depicted in Figure 3.

There are three main categories of Bluetooth LE channel selection algorithms: fixed, dynamic, and adaptive. While fixed channel selection is a straightforward approach, it may not be suitable in environments with high levels of interference. In contrast, dynamic and adaptive channel selection algorithms can adjust to changing interference conditions in real-time, resulting in improved communication quality and reliability [1]. Studies have extensively analyzed the performance of various channel selection algorithms based on metrics such as communication quality, reliability, and energy efficiency. These investigations demonstrate that the selection of a particular algorithm can significantly affect the overall performance of the Bluetooth LE communication [13].

To mitigate the impact of interference and avoid collision, Bluetooth Low Energy uses a frequency hopping function that distributes the transmission signal over several frequencies, thereby reducing the chance of signal collisions.

### 3.1. Random Frequency Hopping

Random frequency hopping is a technique that aims to improve the security of wireless transmissions by preventing interference. It involves rapidly switching between different radio frequencies in a random manner during transmission.

The main concept behind random frequency hopping is that by constantly changing the transmission frequency, it becomes difficult for interfering signals or eavesdroppers to disrupt the communication. This technique is commonly used in military communications and other applications where secure and reliable wireless communication is crucial [14].

Frequency selection in traditional frequency hopping is based on a well-known pseudo-random sequence. As these sequences are fixed, they are not capable of adapting to the current state of the wireless environment. This can lead to frequency overlaps and channel congestion, impacting the signal quality and throughput.

Let $f_{base}$ represent the base frequency of the BLE signal and $f_{offsets}$ denote a set of offset frequencies. The set of valid frequencies for the random frequency hopping scheme is defined as $F=\{f_{base}+f_{offsets}\}$, where + represents the set union operation. The length of *F* is typically limited to a small number of channels (e.g., 37 for BLE).

At each time slot, a random frequency is chosen from *F* and the next hop occurs at the selected frequency. The randomness can be achieved by employing a pseudo-random number generator (PRNG) to obtain a random index in *F*.

Assuming *N* is the length of the hopping sequence, $h_{n}\in F$ represents the frequency used for the nth time slot, 1≤n≤N where the sequence can be defined as:(1)$$h_{n}=f_{base}+f_{offsets_{r(n)}}$$
where r(n) is a pseudo-random number generator (PRNG) that maps values in the range 1,…,|F| to a specific value in *F* for each time slot (n).

The pseudo-random mapping function should be secure and unpredictable to prevent attackers from predicting the hopping sequence. Common examples of PRNGs include the Linear Congruential Generator (LCG) or the Mersenne Twister PRNG [19].

While both LCG and Mersenne Twister generate random-looking sequences of numbers, they differ in various ways, including speed, period length, randomness, and security.

LCG is a straightforward and fast PRNG that generates a sequence of numbers through a linear recurrence relation. It has the following form:(2)$$x_{n+1}=\left(ax_{n}+c\right)\;\mathrm{mod}\;m$$
where $x_{n}$ is the nth generated number, *a* is the multiplier, *c* is the increment, *m* is the modulus, and mod is the modulo operation. LCGs are simple to implement and require only a small amount of memory. However, they have some significant drawbacks, including low randomness and predictability, which can make the frequency hopping sequence vulnerable to attacks.

On the other hand, Mersenne Twister PRNG is a much more sophisticated and secure PRNG with a much longer period length. It is based on a twisted generalized feedback shift register that can generate a sequence of 32-bit integers. The Mersenne Twister PRNG offers better randomness and security than LCG due to its more complex design. The Mersenne Twister PRNG is also more computationally intensive, requiring more memory and processing power, but offers a much longer period length and better statistical randomness. Thus, it is simulated in this research for the frequency assignment of BLE signals.

Using random frequency hopping for frequency selection of BLE signals can be implemented by Algorithm 2.   
**Algorithm 2 ** BLE frequency hopping simulation and performance evaluation.
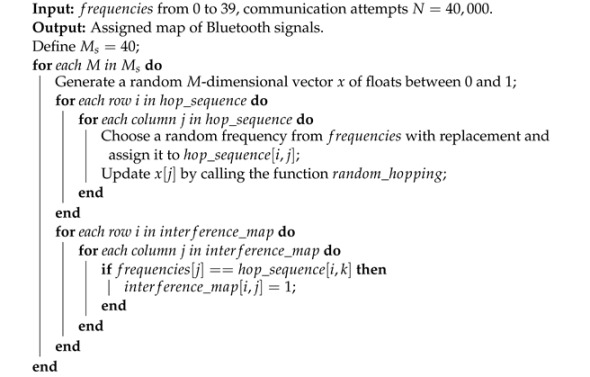


### 3.2. Chaotic Frequency Hopping

Chaotic frequency hopping is a technique that introduces chaos into the frequency hopping pattern to increase the security of wireless communications. This technique utilizes a chaotic map to generate the hopping sequence, which is used to select the frequency channel for each transmission [20].

A chaotic function in frequency hopping refers to a mathematical function that generates a random-like sequence of frequencies. In comparison to the previous pseudo-random sequence, these sequences are dynamic and continuously changing, adapting to the current state of the wireless environment. Chaotic sequences allow BLE devices to hop over previously used frequencies, enabling diversified transmissions and mitigating interference effects.

Various chaotic maps can be used for frequency hopping, including the logistic map, the Hénon map, and the Lorenz system. Each of these maps has different properties and can be used to generate different types of chaotic sequences. A comparison of different chaotic frequency hopping techniques is presented in Table 1.

In general, using chaotic frequency hopping can be a beneficial technique to enhance wireless communication security but it can also pose implementation challenges and increase system complexity. Therefore, it is crucial to carefully weigh the benefits and drawbacks of this approach. However, the logistic map is a simple chaotic map that can effectively generate random sequences, which makes it a potential option for wireless communication systems seeking to implement chaotic frequency hopping.

The logistic map is a mathematical function that is often used as an example of how chaotic behavior can arise from a simple non-linear system. The map is defined by the following equation:(3)$$x_{n+1}=r\cdot x_{n}\cdot \left(1-x_{n}\right)$$
where $x_{n}$ is the current value at time step *n* and *r* is a control parameter that determines the rate of change. The logistic map exhibits chaotic behavior for certain values of *r*, resulting in an unpredictable and aperiodic sequence of values of $x_{n}$.

To apply the logistic map to frequency hopping, we first map the generated values of $x_{n}$ to the frequency range we wish to hop over. This can be accomplished by defining the range of valid frequencies as $F=\{f_{1},f_{2},\ldots ,f_{m}\}$, with *m* being the number of available channels, and mapping the frequency values $x_{n}$ to this valid range by:(4)$$h_{n}=F\left(x_{n}\right)=f_{\left\lfloor \left(x_{n}\times \left(m-1\right)\right)+1\right\rfloor }$$
where $h_{n}$ is the corresponding frequency selected at time step *n* and ⌊⌋ is the floor function used to round the value of $(x_{n}\times \left(m-1\right))+1$ to the nearest integer.

The logistic map chaotic frequency hopping has the benefit of flexibility and adaptiveness, as the hopping sequence is dependent on the current state of the wireless environment. Attackers cannot predict the hopping sequence since it is generated from the chaotic behavior of the logistic map; therefore, it is much more secure than traditional hopping methods.

The logistic map chaotic frequency hopping for frequency selection of BLE signals can be implemented using Algorithm 3.
**Algorithm 3 ** Bluetooth Signal Assignment using Chaotic Frequency Hopping.
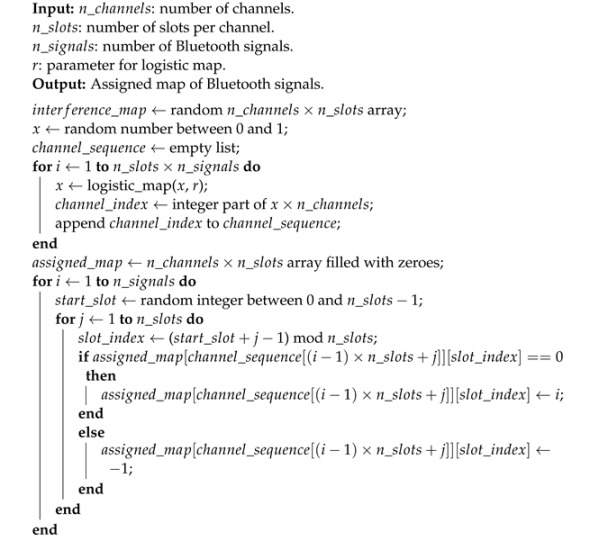


The logistic map has a variety of properties that make it useful in many applications. One example is its ability to generate pseudo-random numbers, which is useful in cryptography and other areas where randomness is important.

In the context of chaotic frequency hopping, the logistic map can be used to create a sequence of pseudo-random numbers that determine the frequency hopping pattern. Using a unique initial condition for each transmission, a sequence of frequencies that appears to be random and unpredictable can be generated, making it difficult for an attacker to intercept the communication.

However, it is important to note that the logistic map also has some limitations. For instance, it can be sensitive to initial conditions, which means that even small changes in the initial state of the system can result in significantly different outcomes. Additionally, the map can exhibit period-doubling bifurcations, which can make the behavior of the system more challenging to anticipate.

### 3.3. Adaptive Frequency Hopping

Adaptive frequency hopping is a technique used in wireless communication systems to dynamically adjust the hopping sequence based on the interference and noise conditions of the channel. The idea is to monitor the channel quality and adapt the hopping sequence to avoid the frequency bands with high interference or noise [21].

The adaptive frequency hopping technique typically involves several steps. First, the quality of the channel is monitored to detect the frequency bands with high interference or noise. This can be performed by measuring the signal-to-interference-plus-noise ratio (SINR) or other quality metrics. Then, a new hopping sequence is selected that avoids those frequency bands. This can be performed by randomly selecting a new sequence that avoids the problematic frequency bands or using a predefined algorithm to generate the new sequence. The new hopping sequence is updated in the system and used for future transmissions. The system continues to monitor the channel quality and adapt the hopping sequence as needed [22].

The advantage of adaptive frequency hopping is that it can improve the reliability and performance of the wireless communication system in the presence of interference and noise. It can also increase the security of the system by making it more difficult for eavesdroppers to intercept the transmitted data. Additionally, the use of the logistic map can simplify the implementation of adaptive frequency hopping by generating a pseudo-random sequence that avoids problematic frequency bands. The disadvantage is that it requires additional hardware and software to implement, and can increase the complexity and cost of the system [23].

The use of adaptive frequency hopping in wireless communication systems is becoming increasingly popular due to its ability to dynamically adjust the hopping sequence based on interference and noise conditions. There are several adaptive frequency hopping techniques that can be used, and a comparison of some of the most common techniques is as follows.

The threshold-based technique involves setting a threshold value for the interference or noise level and changing the hopping sequence if the measured interference or noise level exceeds the threshold. The advantage of this technique is its simplicity and minimal computational resource requirements. However, it may not be effective in all situations, as interference and noise levels can vary widely.

The learning-based technique involves using machine learning algorithms to learn the interference and noise patterns in the channel and adapt the hopping sequence accordingly. The advantage of this technique is that it can adapt to a wide range of interference and noise conditions and improve the performance over time. However, it requires significant computational resources and may not be practical for all systems.

The distributed technique involves using a distributed approach to adapt the hopping sequence, where each node in the network monitors the channel quality and communicates with other nodes to exchange information and adapt the hopping sequence. The advantage of this technique is that it can be more robust and adaptive to changing conditions than centralized approaches. However, it requires more communication and coordination between nodes, which can increase the overhead and complexity of the system.

The hybrid technique combines multiple adaptive frequency hopping techniques, such as threshold-based and learning-based approaches, to improve the overall performance of the system. The advantage of this technique is that it can leverage the strengths of each approach and adapt to a wide range of interference and noise conditions. However, it can be more complex and require more computational resources than single-technique approaches.

The choice of the adaptive frequency hopping technique depends on the specific requirements and constraints of the wireless communication system. Threshold-based techniques may be suitable for simple systems with low interference and noise levels, while learning-based and distributed techniques may be more appropriate for complex systems with high interference and noise levels. Hybrid techniques may offer the best performance and adaptability, but require more resources and complexity to implement.

## 4. Proposed Algorithm

A proposed chaotic frequency hopping technique based on optimizing the update coefficient *r* to minimize interference and ensure zero self-interference among hopping BLE nodes could work as follows:Define a set of *N* hopping frequencies to be used by the BLE nodes.Initialize the update coefficient *r* to a random value between 3.6 and 4.0.For each BLE node, generate a chaotic sequence using the logistic map with the current value of *r*. This sequence will be used to select the hopping frequencies for that node.Calculate the interference matrix *I* of size N×N for the current set of hopping frequencies, where $I_{i,j}=1$ if frequency *i* interferes with frequency *j* and $I_{i,j}=0$ otherwise.For each BLE node, calculate the interference caused by that node on all other nodes using the interference matrix *I*. If the node is interfering with itself, adjust its chaotic sequence to select a different frequency.Update the value of *r* using an optimization algorithm such as gradient descent or simulated annealing to minimize the interference among the nodes. Repeat steps 3–6 until interference is minimized and self-interference is eliminated.

This technique uses chaos theory to generate random and unpredictable hopping sequences for each BLE node, which helps to avoid predictable patterns and reduce interference. By optimizing the update coefficient *r*, the chaotic behavior can be fine-tuned to minimize interference and ensure zero self-interference among the nodes. However, this approach may require significant computational resources and may be more complex to implement than simpler frequency hopping techniques. The proposed algorithm is implemented as depicted in the flowchart in Figure 4.

To compare among the four frequency hopping techniques (random, chaotic, adaptive, and proposed), we use the following criteria:Interference reduction: This technique should reduce the amount of interference among hopping nodes.Self-interference avoidance: This technique should ensure that each hopping node does not interfere with itself.Frequency diversity: This technique should use a wide range of frequencies to avoid congested frequency bands.Synchronization: This technique should ensure synchronization among hopping nodes to avoid collisions.Adaptivity: This technique should adapt to changes in the environment, such as the presence of new devices or interference sources.Complexity: This technique should be simple and easy to implement.

Based on these criteria, we compare the four techniques as follows in Table 2.

From Table 2, we see that the proposed technique offers the highest levels of interference reduction, self-interference avoidance, frequency diversity, synchronization, and adaptivity, while still being relatively simple to implement. The chaotic and adaptive techniques also offer high levels of interference reduction and frequency diversity, but they may be more complex to implement. The random technique is the simplest to implement but offers low levels of interference reduction and frequency diversity.

## 5. Simulation Results

To simulate the effect of Bluetooth interference on Wi-Fi signals, Bluetooth and Wi-Fi signals are simulated according to the following parameters.

First, set up the Wi-Fi signal simulation parameters, including the number of packets = 1000, the number of bits per packet = 1000 and SNR values ranging from 0 to 30 dB. The Wi-Fi carrier frequencies are chosen among the three mentioned non-overlapping channels, (2412, 2437, 2462) each with 20 MHz bandwidth. The first carrier frequency is chosen for simulation. Then, the Bluetooth signal is simulated with a hopping carrier frequency within the 10 available channels to coexist with the simulated Wi-Fi signal with 2 MHz bandwidth. The Bluetooth packets are simulated via 1600 time slots per packet.

For the effect of interference on the bit error rate or packet error rate of a Wi-Fi signal we used an AWGN channel model and Monte Carlo simulation to estimate the error rates for different levels of interference as a function of the Bluetooth carrier frequency location within the Wi-Fi signal bandwidth, as well as the interference level and duration with respect to the Wi-Fi packet. The error vector magnitude (EVM) is a measure of how accurately the demodulated signal matches the original transmitted signal. It is typically expressed as a percentage or in decibels (dBs).

A high EVM indicates that the demodulated signal has a high level of distortion or error compared to the original transmitted signal. This can lead to increased bit error rates and a reduced system performance. A low EVM, on the other hand, indicates that the demodulated signal closely matches the original transmitted signal and that the system is performing well.

Figure 5 depicts the effect of the Bluetooth carrier frequency location within the Wi-Fi signal bandwidth on the Wi-Fi signal in terms of the EVM.

As the carrier frequency of the interfering BLE signal is near to the Wi-Fi carrier frequency, the EVM increases and vice versa. If the BLE carrier frequency is equal to the Wi-Fi carrier frequency, the EVM = 0 dB, indicating a total Wi-Fi packet loss. Thus, in the upcoming simulations, an interference is only considered if the BLE and Wi-Fi signals have the same carrier frequency.

Frequency hopping techniques for BLE signals are simulated for minimal interference with Wi-Fi signals; the simulation setup is illustrated in Algorithm 4.
**Algorithm 4 ** BLE frequency hopping simulation and performance evaluation.
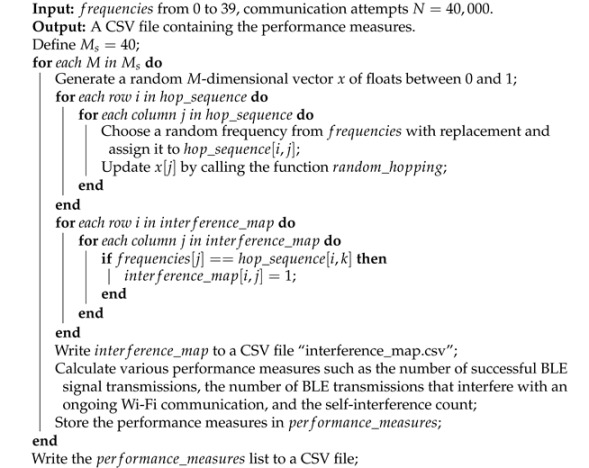


Different types of frequency hopping algorithms represented in earlier sections are compared with the proposed algorithm to compare the existing algorithms in terms of interference avoidance and success rate, in addition to the computational complexity. In this section, we presented the results of our simulations for the different types of frequency hopping techniques that we studied, including random, chaotic, adaptive random, and adaptive chaotic. We also proposed an optimized chaotic frequency hopping technique that outperforms the other techniques in terms of reducing interference with Wi-Fi signals and achieving a high success rate for connecting Bluetooth nodes.

We conducted simulations in an environment that includes existing interference from Wi-Fi signals and interfering Bluetooth nodes. We compared several performance metrics, including the total interference rate, total successful connection rate, and trail execution time for channel selection processing time. These metrics were plotted for different numbers of accessing Bluetooth nodes.

The total interference rates for different frequency hopping techniques are depicted in Figure 6.

It is obviously observed that the total interference rate increases as the number of accessing Bluetooth nodes increases for all frequency hopping techniques. The chaotic frequency hopping technique had the highest total interference rate compared to the other techniques, while the adaptive frequency hopping technique had a lower interference rate than both the chaotic and random techniques, as shown in Figure 6. However, our proposed optimized chaotic frequency hopping technique had a slightly higher interference than that of the adaptive technique for small numbers of Bluetooth nodes. As the number of Bluetooth nodes increased, our proposed technique had a much lower interference than the adaptive technique.

The total successful connection rate was defined as the number of successfully assigned frequencies to the Bluetooth nodes without interfering with the Wi-Fi signals. The total successful connection rate comparison for different frequency hopping techniques is shown in Figure 7.

It is clear that the total successful connection rate increases as the number of accessing Bluetooth nodes increases for all frequency hopping techniques. The chaotic frequency hopping technique had the lowest success rate compared to the other techniques, while the adaptive frequency hopping technique had a higher success rate than both chaotic and random techniques, as shown in Figure 7. However, the proposed optimized chaotic frequency hopping technique demonstrated a superior performance compared to all other techniques, including the adaptive technique, particularly when the number of Bluetooth nodes was large. However, for a node count of up to 20, the proposed technique did not outperform the adaptive technique.

The Mersenne Twister PRNG and the logistic map have different computational complexities, which can be compared using Big O calculations.

The Mersenne Twister PRNG has a time complexity of O(n) for generating n pseudo-random numbers. This complexity is due to the initialization process that sets the initial state of the PRNG. Once the initialization is complete, generating each number takes a constant time. Therefore, the computational complexity of generating N random numbers using the Mersenne Twister is O(N), i.e., linear time, while the logistic map is a nonlinear function in which the current value is calculated based on the previous value. The time complexity of a single iteration of the logistic map is O(1), as it only involves basic arithmetic operations.

The trial execution time is also measured for the channel selection processing time. The trial execution time comparison for different frequency hopping techniques is shown in Figure 8.

It is found that the chaotic frequency hopping technique had the lowest execution time, demonstrating the net advantage of chaotic techniques due to its simplicity, as shown in Figure 8. The adaptive techniques required much longer execution times, with the adaptive random technique requiring the highest among all. The proposed technique required a very close time to that of the conventional chaotic technique, which was much shorter than that of all other techniques.

Overall, our results demonstrated that the proposed optimized chaotic frequency hopping technique achieved a better balance between reducing interference with Wi-Fi signals, achieving a high success rate for connecting Bluetooth nodes, and requiring minimal trial execution times, making it a suitable technique for managing interference in wireless communication systems.

It is worth mentioning that both adaptive techniques have the same interference and success rates, while the adaptive chaotic hopping requires notably less execution time than that of the adaptive random technique.

## 6. Conclusions

In this study, we proposed an optimized chaotic frequency hopping technique for managing interference in wireless communication systems. We compared this technique with several other frequency hopping techniques, including random, chaotic, adaptive random, and adaptive chaotic. Our simulations were conducted in an environment that included existing interference from Wi-Fi signals and interfering Bluetooth nodes. Our results showed that the total interference rate increases as the number of accessing Bluetooth nodes increases for all frequency hopping techniques. The chaotic frequency hopping technique had the highest total interference rate compared to the other techniques, while the adaptive frequency hopping technique had a lower interference rate than both the chaotic and random techniques. However, our proposed optimized chaotic frequency hopping technique had a slightly higher interference than that of the adaptive technique for small numbers of Bluetooth nodes. As the number of Bluetooth nodes increased, our proposed technique had a much lower interference than the adaptive technique.

We also found that the total successful connection rate increases as the number of accessing Bluetooth nodes increases for all frequency hopping techniques. The chaotic frequency hopping technique had the lowest success rate compared to the other techniques, while the adaptive frequency hopping technique had a higher success rate than both chaotic and random techniques. However, our proposed optimized chaotic frequency hopping technique outperformed all other techniques, including the adaptive technique, especially at large numbers of Bluetooth nodes.

Finally, we measured the trial execution time for the channel selection processing time. We found that the chaotic frequency hopping technique had the lowest execution time, demonstrating the net advantage of chaotic techniques due to its simplicity. The adaptive techniques required much longer execution times, with the adaptive random technique requiring the highest among all. The proposed technique required a very close time to that of the conventional chaotic technique, which was much shorter than that of all other techniques. Overall, our results demonstrated that the proposed optimized chaotic frequency hopping technique achieves a better balance between reducing interference with Wi-Fi signals, achieving a high success rate for connecting Bluetooth nodes, and requiring minimal trial execution time. This makes it a suitable technique for managing interference in wireless communication systems.

The paper makes several contributions in the field of wireless communication systems. First, it proposes a new optimized chaotic frequency hopping technique that minimizes interference with Wi-Fi signals while achieving a high success rate for connecting Bluetooth nodes. Second, it compares this technique with existing random, chaotic, and adaptive frequency hopping techniques using several performance metrics, such as interference rate, successful connection rate, and trial execution time. Third, it demonstrates through simulation that the proposed technique outperforms all other techniques, especially at large numbers of Bluetooth nodes. Fourth, it shows that the proposed technique requires minimal trial execution time, making it a suitable solution for managing interference in wireless communication systems. Overall, the paper contributes new knowledge and techniques that can enhance the performance of wireless communication systems in real-world environments with existing interference from other wireless signals.

## Figures and Tables

**Figure 1 sensors-23-05183-f001:**
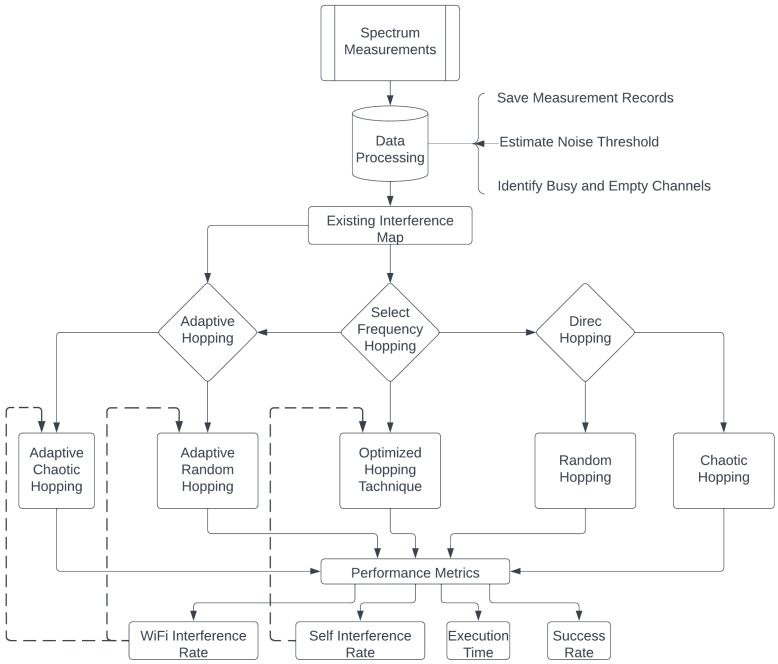
Research methodology.

**Figure 2 sensors-23-05183-f002:**
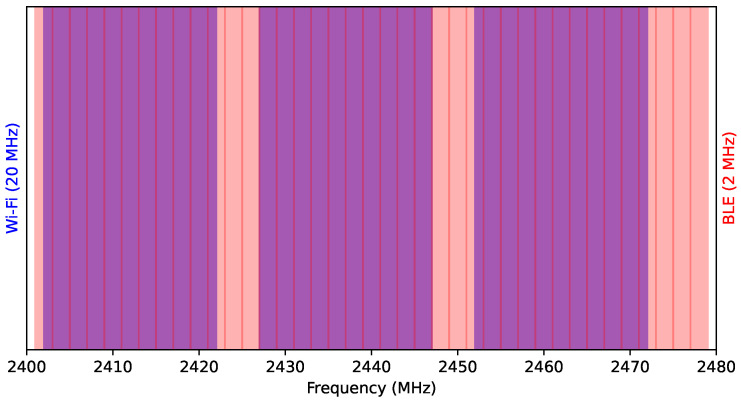
Frequency Plan for 2.4 GHz ISM band.

**Figure 3 sensors-23-05183-f003:**
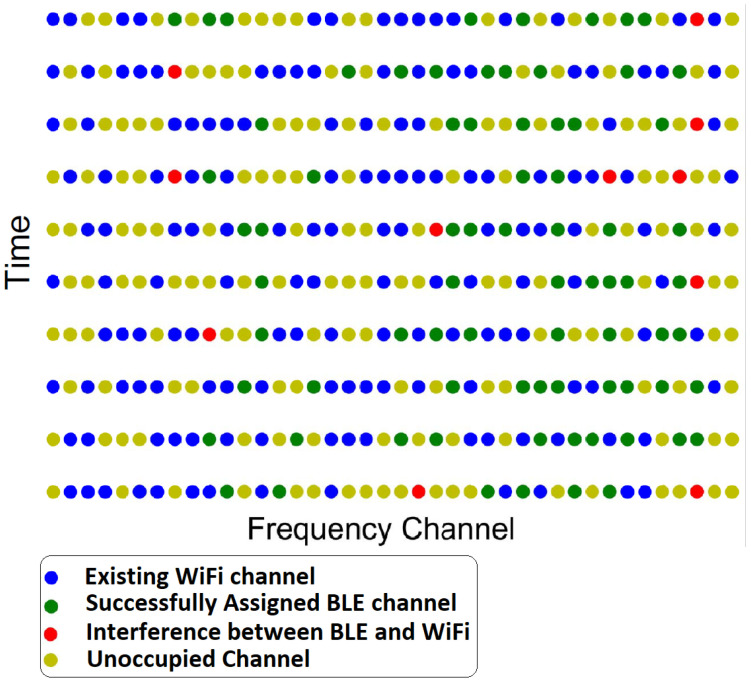
BLE Frequency Hopping to Coexist with Wi-Fi.

**Figure 4 sensors-23-05183-f004:**
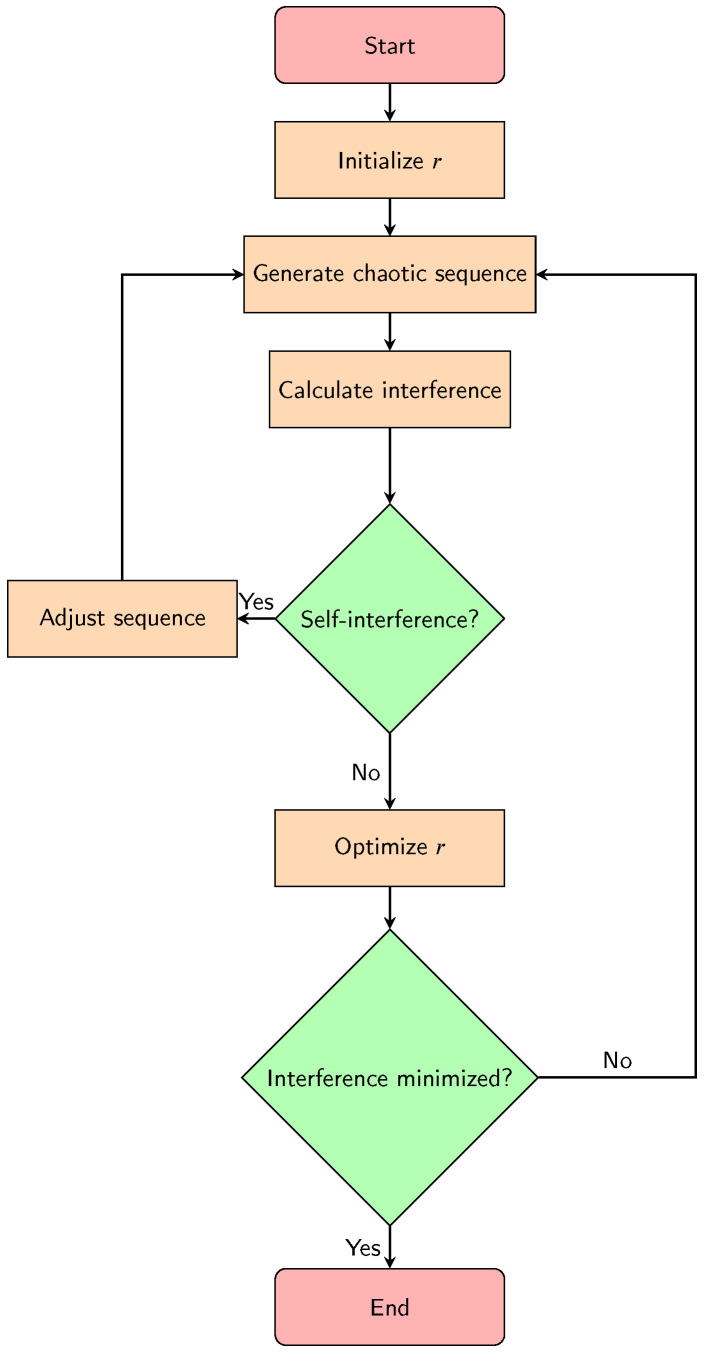
Flowchart of the proposed chaotic frequency hopping technique.

**Figure 5 sensors-23-05183-f005:**
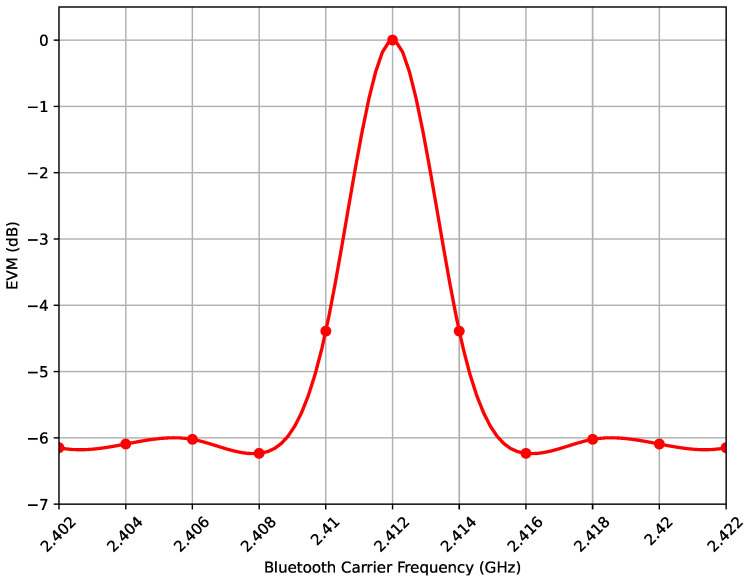
BLE Frequency Hopping to Coexist with Wi-Fi.

**Figure 6 sensors-23-05183-f006:**
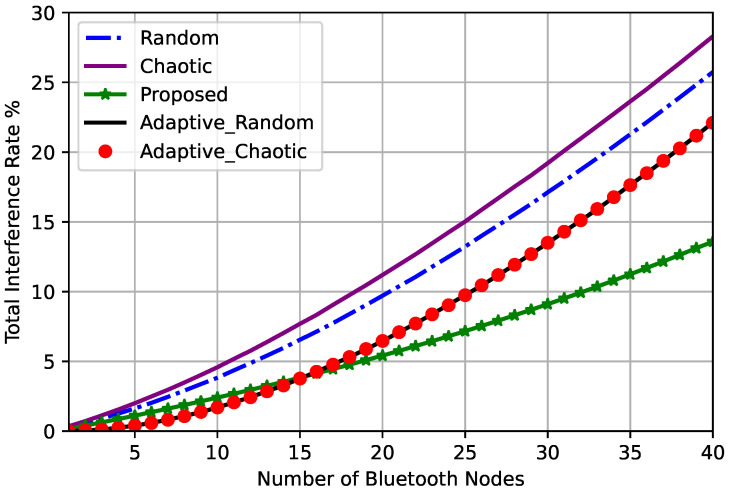
Total interference rates for different frequency hopping techniques.

**Figure 7 sensors-23-05183-f007:**
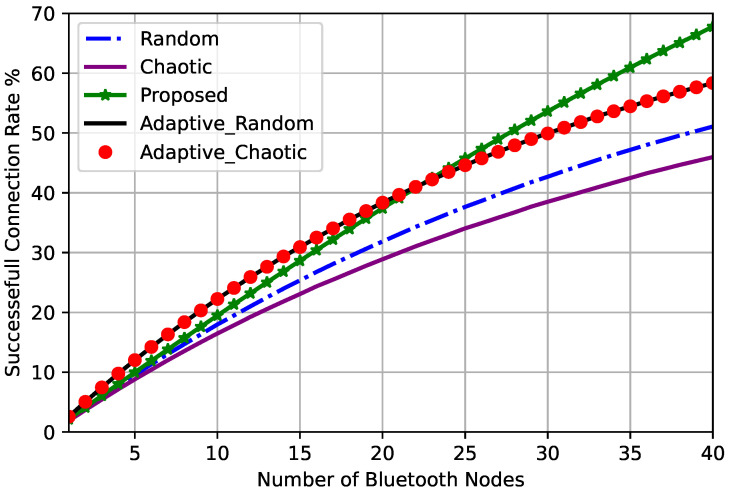
Total successful connection rates for different frequency hopping techniques.

**Figure 8 sensors-23-05183-f008:**
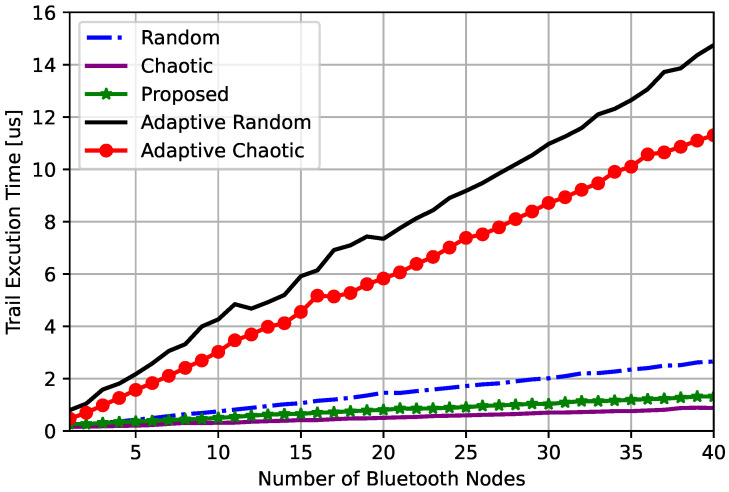
Trial execution time comparison for frequency hopping techniques.

**Table 1 sensors-23-05183-t001:** Comparison of Chaotic Frequency Hopping Techniques.

Technique	Advantages	Disadvantages	Complexity
Logistic Map	Easy to implement	Sensitivity to initial conditions	Low
Hénon Map	High degree of randomness	Complex to implement	Moderate
Lorenz System	Highly complex behavior	Difficult to implement	High
Hybrid Techniques	High degree of randomness and security	Complex to implement	High

**Table 2 sensors-23-05183-t002:** Comparison of frequency hopping techniques.

Criteria	Random	Chaotic	Adaptive	Proposed
Interference reduction	Moderate	Low	High	High
Self-interference avoidance	Low	Low	High	High
Frequency diversity	Moderate	High	High	High
Synchronization	Low	Low	Moderate	High
Adaptivity	Low	Low	High	High
Complexity	Moderate	Simple	Complex	Moderate

## Data Availability

Not applicable.

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
