# Peer review of "Improved Spectrum Coexistence in 2.4 GHz ISM Band Using Optimized Chaotic Frequency Hopping for Wi-Fi and Bluetooth Signals"

_sensors, 2023, doi:10.3390/s23115183_

Round 1
Reviewer 1 Report
The article is well-written and the result improves the state-of-the-art on coexistence of Bluetooth and Wi-Fi signals with reduced interference.
Below are comments that need to be addressed regarding presentation and clarity of information:
1) Would be better to add reference(s) for justification of the statements in the 2nd paragraph of Introduction, lines 32-36.
2) What is mentioned in lines 43-45 is cancelled in the next paragraph. Please re-elaborate for clarity.
3) Except the last sentence of the last paragraph of the Introduction, all the rest should be deleted because it is repetition from what is written earlier.
4) Define FHSS abbreviation first time in line 119.
5) Elaborate more on figure 3. Which are the vacant unoccupied channels on the figure?
6) Rewrite the sentence at line 375 for better language.
7) For lines 409-411, the proposed technique did not outperform the adaptive technique for a node number of up to 20+. Rewrite this sentence.
The English language is generally very good. Some of the review comments are about improving it in few places.
Reviewer 2 Report
this article proposes an optimized chaotic frequency hopping technique for managing interference between Wi-Fi and Bluetooth Low Energy (BLE) signals in wireless communication systems. The proposed technique achieves a better balance between reducing interference with Wi-Fi signals, achieving a high success rate for connecting BLE nodes, and requiring minimal trial execution time.
The evaluation of the proposed method is based on simulations, and the results demonstrate that the proposed technique is effective in managing interference in wireless communication systems.
But there are still some issues:
1. This article does not provide a detailed explanation of the proposed optimized chaotic frequency hopping technology.
Readers may need more information to understand the working principle of this technology and its differences from traditional frequency hopping methods.
And there is no comparison with other existing wireless communication system interference management technologies,
which makes it difficult for readers to evaluate the effectiveness of the proposed technology.
2. The evaluation of the proposed method is only based on simulation, without actual testing,
and the sample size used in the simulation is relatively small, which may limit the universality of the results.
Therefore, readers may question whether this method is effective in practical applications and can be applied to other wireless communication systems.
3. This article only focuses on the spectrum coexistence problem between Wi Fi and BLE signals,
and does not consider the coexistence problem between other wireless communication systems.
Therefore, readers may need more information to understand whether this technology can be applied to the coexistence problem between other wireless communication systems.
4. This article does not explicitly explain how the proposed technology achieves a better balance between reducing interference with Wi Fi signals and achieving high success rates in BLE node connections. Readers may need more information to understand how this balance is achieved.
5. This article does not discuss the potential impact of the proposed technology on power consumption or battery life in wireless communication systems. This information may be important for readers interested in implementing the proposed technology in practical applications.
this article proposes an optimized chaotic frequency hopping technique for managing interference between Wi-Fi and Bluetooth Low Energy (BLE) signals in wireless communication systems. The proposed technique achieves a better balance between reducing interference with Wi-Fi signals, achieving a high success rate for connecting BLE nodes, and requiring minimal trial execution time.
The evaluation of the proposed method is based on simulations, and the results demonstrate that the proposed technique is effective in managing interference in wireless communication systems.
But there are still some issues:
1. This article does not provide a detailed explanation of the proposed optimized chaotic frequency hopping technology.
Readers may need more information to understand the working principle of this technology and its differences from traditional frequency hopping methods.
And there is no comparison with other existing wireless communication system interference management technologies,
which makes it difficult for readers to evaluate the effectiveness of the proposed technology.
2. The evaluation of the proposed method is only based on simulation, without actual testing,
and the sample size used in the simulation is relatively small, which may limit the universality of the results.
Therefore, readers may question whether this method is effective in practical applications and can be applied to other wireless communication systems.
3. This article only focuses on the spectrum coexistence problem between Wi Fi and BLE signals,
and does not consider the coexistence problem between other wireless communication systems.
Therefore, readers may need more information to understand whether this technology can be applied to the coexistence problem between other wireless communication systems.
4. This article does not explicitly explain how the proposed technology achieves a better balance between reducing interference with Wi Fi signals and achieving high success rates in BLE node connections. Readers may need more information to understand how this balance is achieved.
5. This article does not discuss the potential impact of the proposed technology on power consumption or battery life in wireless communication systems. This information may be important for readers interested in implementing the proposed technology in practical applications.
